# Au-TiO_2_-Coated Spectroscopy-Based Human Teeth Disorder Detection Sensor: Design and Quantitative Analysis

**DOI:** 10.3390/mi14061191

**Published:** 2023-06-02

**Authors:** Sumaiya Akhtar Mitu, Kawsar Ahmed, Francis M. Bui, Li Chen, Lassaad K. Smirani, Shobhit K. Patel, Vishal Sorathiya

**Affiliations:** 1Department of Information Technology, University of Information Technology & Sciences (UITS), Dhaka 1212, Bangladesh; mitu.ict12@gmail.com; 2Group of Biophotomatiχ, Department of Information and Communication Technology (ICT), Mawlana Bhashani Science and Technology University (MBSTU), Tangail 1902, Bangladesh; 3Department of Electrical and Computer Engineering, University of Saskatchewan, Saskatoon, SK S7N 5A9, Canada; francis.bui@usask.ca (F.M.B.); lic900@usask.ca (L.C.); 4The Deanship of Information Technology and E-learning, Umm Al-Qura University, Mecca 24382, Saudi Arabia; lksmirani@uqu.edu.sa; 5Computer Engineering Department, Marwadi University, Rajkot 360003, India; shobhitkumar.patel@marwadieducation.edu.in; 6Faculty of Engineering and Technology, Parul Institute of Engineering and Technology, Parul University, Vadodara 391760, India; vishal.sorathiya9@gmail.com

**Keywords:** aqueous solution, dental disorders, spectroscopy, SPR sensor, teeth disorder detection, quantitative analysis

## Abstract

Human tooth functionality is the most important for the human body to become fit and healthy. Due to the disease attacks in human teeth, parts may lead to different fatal diseases. A spectroscopy-based photonic crystal fiber (PCF) sensor was simulated and numerically analyzed for the detection of dental disorders in the human body. In this sensor structure, SF11 is used as the base material, gold (Au) is used as the plasmonic material, and TiO_2_ is used within the gold and sensing analyte layer, and the sensing medium for the analysis of the teeth parts is the aqueous solution. The maximum optical parameter values for the human tooth parts enamel, dentine, and cementum in terms of wavelength sensitivity and confinement loss were obtained as 28,948.69 nm/RIU and 0.00015 dB/m for enamel, 33,684.99 nm/RIU and 0.00028 dB/m, and 38,396.56 nm/RIU and 0.00087 dB/m, respectively. The sensor is more precisely defined by these high responses. The PCF-based sensor for tooth disorder detection is a relatively recent development. Due to its design flexibility, robustness, and wide bandwidth, its application area has been spreading out. The offered sensor can be used in the biological sensing area to identify problems with human teeth.

## 1. Introduction

Teeth are an important part of the human body. A healthy and bright smile mostly depends on strong teeth. Basically, the human tooth can be divided into two parts: the crown and the root. The crown is the external part that is visible, but the root extends below, so it cannot be seen. These teeth consist of four types of tissues, (i) of which enamel (En) is the strongest among the other tissues. It is formulated of phosphorus, calcium, and hydroxyapatite, (ii) dentin (De) is the calcified part and looks similar to a bone; (iii) cementum (Ce) is the softer part than De and En, and it is covered by bones and gums, and (iv) pulp is the core part that connects the teeth with nerves, blood vessels, and other soft tissues [1,2]. This part is used to send signals. People with strong and healthy teeth can enjoy any kind of snack or food without any difficulties. On the other hand, researchers have found that unhealthy teeth may lead to different problems such as bad breath, and tooth disease is also harmful to diabetic patients [3].

There are some existing methods to detect diseases in human teeth, such as diagnodent, fiber optic transillumination (FOTI), electrical caries-monitor (ECM), quantitative light induced fluorescence (QLF), digital imaging fiber-optic transillumination (DIFOTI), digital radiographs, and so on [4]. These detection methods have some disadvantages such as tissue width, temperature, and hydration. The teeth of the human body can be used as the sensing medium. For this reason, a spectroscopy-based sensor was analyzed for detecting tooth diseases. There are a huge number of advantages to spectroscopy-based sensors, such as the concentration of different sensing analytes that can be examined within a solution. In addition, the quality of the solution can also be examined, and it is more accessible than other chromatography analyses.

Through spectroscopy-based analysis, different analytes have been used as the sensing material. At present, spectroscopy is the most straightforward, cost-effective, robust, and flexible sensing technique for disease detection. The light absorption technique transfers the energy photon to the atoms of a medium. Next, the energy is converted into other forms. This process creates the power and transmitted spectrum to identify the disease. Different PCF-based sensors have been developed for calculating sensing performance for different materials. In the PCF-based sensing technology, various sensing techniques were discussed and developed by the researchers. Sensing techniques based on Mach Zehnder interferometer, fiber grating, spectroscopy, and wavelength division multiplexing are the most demanding approaches. The optical light intensity has been used for prominent sensing techniques such as colorimetric [5], light scattering [6], surface plasmon resonance (SPR) sensors [7], scattering sensors [8], electrochemiluminescence sensors [9], fluorescent sensors [10], and so on. Various analyses were performed, and the relative sensitivity of 45% was noted for identifying hazardous chemicals that are harmful to the human body [11]. Moreover, spiral-based PCF (SPCF) was introduced for detecting the gas sensor and pointed to a relative sensitivity of 55% [12]. To increase the relative sensitivity (RS), different techniques were invented. In addition, RS of 93.5%, as well as the confinement loss of 0.004 dB/m, was gained for the lattice square PCF-based sensor structure. Recently, the hollow core PCF (HC-PCF) was designed for identifying the blood components in the human body and obtained a maximum RS of 94.03% [13].

In the proposed work, a novel human teeth disorder detection sensor is designed and numerically analyzed. Dentine, enamel, and cementum from human teeth are employed as the sensor’s detecting components. Here, a large range of fiber optic properties, including birefringence, coupling length, numerical aperture, and others, are examined.

## 2. Design and Methodology

A cross-sectional view of the simulated sensor structure for human teeth disease detection and the corresponding orthogonal core modes at the wavelength 800 nm are shown in Figure 1. The proposed sensor structure consists of two-layer air hole variations. There are two types of air holes. The large air hole diameters are 2 µm and the small air hole diameters are 1 µm. The distance between the centers of the air holes is known as the pitch difference. The value of pitch is also kept at 3.5 µm. Generally, silica is used as a base material but silica has a lower refractive index (RI) than the tooth parts. For these reasons, a higher RI containing material SF11 was used as a base material in this work. SF11 [14] has a higher RI than the tooth parts of enamel, dentine, and cementum. The gold (Au) layer is coated on the sensor surface so that plasmons can be created at the interface of the gold and analyte layer. Moreover, another small layer of TiO_2_ is used between the Au layer and analyte layer to remove the adhesive problem. The thickness of the gold layer, TiO_2_ layer, and analyte layer were kept at 40 nm, 10 nm, and 0.7 µm, respectively. The perfectly matched layer (PML) can be applied as the boundary condition of the PCF sensor. An outer layer thickness of the PML layer is kept at 1 µm. This PML layer can prevent the absorption of energy from the sensor structure. All these simulation parts were analyzed through the Comsol Multiphysics Software (Version 5.5). The light-propagating core modes can be obtained through the simulated software. The finite element method (FEM) was used to observe the numerical calculations. The optical parameters values such as effective RI, effective area (EA), effective material loss (EML), and confinement loss (CL) can be obtained through the simulated software using the FEM methods [15].

## 3. Fabrication

The analyzed teeth were collected from the human body. After extraction of these teeth, parts were fixed within 4% formaldehyde. Eventually, the teeth were kept in Morse’s solution or 17% EDTA for decalcification over the time period of 28 days. The expanse of decalcification was performed using radiography. After that, the aqueous solution for the teeth parts was kept as the sensing analyte within the simulated structure. The values of the air hole diameters, pitch, and plasmonic 3-layer thickness are the optimized values for the proposed sensor structure. Various material coatings methods such as radio frequency sputtering (RF sputtering), thermal evaporation, and wet chemical deposition were used to coat the external thin layers [1]. There exist distinct metal coating tests to obtain the consistency of the layer thickness. Chemical vapor deposition (CVD) is an impressive method for coating nanometer-thick metal [16]. In addition, a selective filling approach [17] is used to fill the analyte layer that is kept outside the Au and TiO_2_ layers. The recommended experimental setup diagram of the designed sensor structure is visualized in Figure 2. Input laser light hits the sensor structure and the reflected light is collected through the polarizer. From the photo-detector, the optical responses can be visualized.

## 4. Numerical Analysis

### 4.1. RI of SF11

For sensing purposes, different materials are used as the base material, such as BK7, Ag, MoSe_2_, TiO_2_, C, SF10, SF11, and so on [18]. Among all of the materials, SF11 was used as background material in this simulation due to its high refractive index (RI) value. The RI of *SF11* can be measured using the Sellmeier Equation (Equation 1).
(1)$$n_{SF11}=\sqrt{\left(1+\frac{\alpha_{1}\lambda^{2}}{\lambda^{2}-\beta_{1}}+\frac{\alpha_{2}\lambda^{2}}{\lambda^{2}-\beta_{2}}+\frac{\alpha_{3}\lambda^{2}}{\lambda^{2}-\beta_{3}}\right)}$$

Here, $\alpha_{1}$, $\alpha_{2}$, $\alpha_{3}$, $\beta_{1}$, $\beta_{2}$, and $\beta_{3}$ are the Sellmeier constants and the values of the constants are listed below in Table 1.

### 4.2. RI of Au

Plasmonic materials are metals that express negative permittivity. The most commonly used plasmonic materials are silver (Ag) and gold (Au). Moreover, many other materials, such as TiO_2_ and Ta_2_O_5_, show metal-like optical properties in the specific wavelength range. To create plasmonic materials with lower losses and tunable optical characteristics, numerous research teams are investigating various methods. The refractive index of Au can be determined using the Drude–Lorentz Equation (Equation 2).
(2)$$\epsilon_{Au}=\epsilon_{\alpha }-\frac{{\left(\omega_{D}\right)}^{2}}{\omega (\omega +j\gamma_{D})}-\frac{\Delta_{\epsilon }{\left(\Omega_{L}\right)}^{2}}{(\omega^{2}-{\left(\Omega_{L}\right)}^{2}-j\tau_{L}\omega }$$
Here, the dielectric constant of gold $\epsilon_{Au}$ can be represented through the values of angular frequency $\omega =\frac{2\pi c}{\lambda }$, damping frequency $\gamma_{D}=15.92$ THz × 2π, permittivity $\epsilon_{\alpha }=5.9673$, plasmonic frequency $\omega_{D}=2113.6$ THz × 2π, and the weighting factor $\Delta_{\epsilon }=1.09$. The values of the constants used in Equation (Equation 2) are taken from [19].

### 4.3. RI of $TiO_{2}$

An extra layer of titanium oxide (TiO_2_) is used to raise the interaction between the gold layer and the analyte layer. The surface plasmon oscillations occur at the interface of the Au and dielectric due to the variations of charge density [20]. These oscillations are responsible for creating surface plasmon polariton (SPP). This occurs when the sensing layer is under the influence of total internal reflection (TIR). The wavelength-dependent RI of TiO_2_ can be expressed by Equation (Equation 3) [7].
(3)$$n_{TiO_{2}}=\sqrt{5.913+\frac{2.441\times 10^{7}}{\left(\lambda^{2}-0.803\times 10^{7}\right)}}$$
Here, λ is the operating wavelength.

### 4.4. Birefringence

The birefringence is found when the incident ray of the input light splits into two directions for two distinct polarizations: one is ordinary and another is extraordinary polarization [21]. These rays are linear (plane) and circularly polarized modes. In the PCF-based sensor structure, birefringence is an important optical parameter that has a great impact on the applications of polarization maintaining properties [20]. The birefringence of the proposed sensor can be defined by the following Equation (Equation 4) [22].
(4)$$B_{i}\left(RIU\right)=\left|n_{eff}^{X}-n_{eff}^{Y}\right|$$

Here, $n_{eff}^{X}$ and $n_{eff}^{Y}$ are the RI values for the X polarization and Y polarization core modes, respectively.

### 4.5. Beat Length (LB)

The beat length (LB) can be obtained when the phase delay reaches 2 Π, and the state of polarization is identical to the propagation of the input light. This condition can be attained after reaching the propagation length of LB. Beat length is inversely proportional to the birefringence value. This wavelength-dependent optical parameter has a low beat length with the phase value of the sensor structure. Equation (Equation 5) is used to calculate the beat length of the PCF sensor [23].
(5)$$L_{B}\left(nm\right)=\frac{\lambda }{|n_{eff}^{X}-n_{eff}^{Y}|}=\frac{\lambda }{B_{i}}$$
Here, λ is the operating wavelength and $B_{i}$ indicates the polarization-dependent birefringence value.

### 4.6. Coupling Length (LC)

The utmost length of light emanation within the waveguide can be referred to as the coupling length. The propagation energy is transferred from one waveguide to another. The values of coupling length can be achieved using Equation (Equation 6) [24].
(6)$$L_{C}\left(nm\right)=\frac{\lambda }{2\times B_{i}}$$

### 4.7. Numerical Aperture (NA)

The maximum angle that an incident ray can make with regard to the fiber axis is known as the NA of the fiber. It may be determined from the RI difference between the core and the cladding with the following relation of Equation (Equation 7) [25].
(7)$$NA=\sqrt{n_{core}^{2}-n_{cladding}^{2}}$$
Here, $n_{core}$ and $n_{cladding}$ are the effective RI values for the light-propagating core and cladding regions. In this simulation, $SF_{11}$ was used as the core material, and sensing analytes of human teeth parts were applied as the cladding materials.

### 4.8. Power Spectrum

The output power spectrum of a PCF sensor can be defined as the distribution of total power or energy of the input signal within the frequency domain. The output power spectrum can be obtained through the sinusoidal curve of range 0 dB to 1 dB saturation value. The values of this optical parameter can be calculated using Equation (Equation 8) [26].
(8)$$P_{Out}\left(dB\right)=sin{\left(\frac{B_{i}\times \Pi \times L}{\lambda }\right)}^{2}$$
Here, $B_{i}$ and L indicate the birefringence value and the fiber length of the proposed PCF sensor with respect to the variation of operating wavelength λ.

### 4.9. Transmission Spectrum

The transmission spectrum of the sensor is the power-dependent optical parameter. The variations of the output power spectrum with respect to the input power of 0.985945 dB were analyzed through the transmission spectrum. The transmitted power spectrum of the offered sensor may be measured through Equation (Equation 9) [27].
(9)$$T\left(dB/cm\right)=10\times \log\left(\frac{P_{out}}{P_{in}}\right)$$Here, $P_{out}$ and $P_{in}$ are the output power and the applied power value for the simulated sensor structure.

### 4.10. Effective Mode Area (EMA)

The EMA is an expanse of great significance. The EMA can be attained through the integration of the core power and total power value. A low value of EMA grants a high-density power value for the nonlinear effect of the sensor [28]. Moreover, the EMA has the importance of the context of the numerical aperture, wavelength sensitivity, and confinement loss [29,30]. The values of EMA can be determined using Equation (Equation 10).
(10)$$A_{eff}\left(\mu m^{2}\right)=\frac{{\left(\;\int \int \;|E\right|}^{2}{dxdy)}^{2}}{{\;\int \int \;|E|}^{4}dxdy}$$
Here, *E* defines the amplitude of the electric field in the planar medium for the sensor structure.

### 4.11. Wavelength Sensitivity

The wavelength sensitivity (WS) response indicates the summary of the designed sensor performance. The values of the WS responses can be gained from the spectral peak differences of the transmitted power spectrum. The maximum value of sensing performance indicates the effectiveness of the sensor. The WS can be measured using Equation (Equation 11) [31].
(11)$$WS\left(nm/RIU\right)=\frac{\Delta \lambda_{peak}}{\Delta n_{a}}$$
Here, $\Delta \lambda_{peak}$ indicates the spectral peak differences of the transmitted power spectrum and $\Delta n_{a}$ refers to the value of corresponding RI differences.

### 4.12. Confinement Loss (CL)

Confinement loss is the optical measurement of the scattering of the polarized light. Some of the optical light scatters when reflecting from the sensor structure. This portion of the light intensity may affect the sensor and lead to weakness. The CL of the recommended sensor structure can be measured using Equation (Equation 12) [32].
(12)$$CL\left(dB/cm\right)=8.686\times k_{0}\times Im\left(n_{eff}\right)\times 10^{4}$$
Here, $Im(n_{eff})$ denotes the imaginary part of the core mode for the proposed sensor structure.

## 5. Result Analysis and Discussion

### 5.1. Effective Refractive Index (RI)

The RI variation with respect to the operating wavelength is noted in Figure 3 for the orthogonal modes (a) and (b) for X and Y polarizations, respectively. The light-propagated core mode refractive indices for the human tooth parts dentin (De), cementum (Ce), and enamel (Ee) are specified using the red, blue, and green lines. These values are decreasing with the shift of the operating wavelength from 700 nm to 1000 nm. In addition, the RI of the base material SF11 is also shifting downwards with the increase of wavelength from 700 nm to 1000 nm, which is shown in Figure 3a with the violet line. Here, the RI value of SF11 is greater than the tooth parts, which will help the propagated light to pass through the core region as well as obtain a good sensing performance. These effective RI values are the important features for calculating the other optical parameters such as birefringence (Bi), numerical aperture (NA), beat length (LB), coupling length (LC), total effective power (P), and the transmission spectrum (T). All these parameters highly affect the sensing performance of the proposed sensor structure.

### 5.2. Birefringence (Bi)

The value of birefringence can be calculated through the difference of the effective RI values for the two orthogonal polarized modes using Equation (Equation 4). The birefringence values for the human teeth parts are given in Figure 4. The value of Bi is rising gradually with the change of working wavelength. The minimum values of $2.27\times 10^{-5}$, $2.07\times 10^{-5}$, and $1.89\times 10^{-5}$ RIU are noted at the operating wavelength 700 nm for the analyte of human teeth parts, respectively. Moreover, the maximum values of $6.61\times 10^{-5}$, $6.23\times 10^{-5}$ and $5.67\times 10^{-5}$ RIU are noted at the operating wavelength 700 nm for the analyte of human teeth parts, respectively. It is noted that the birefringence value is near the order $10^{-5}$. These values are highly affected by the values of the effective RI, and the value of birefringence is responsible for calculating the value of LB and LC [33].

### 5.3. Beat Length (LB)

The beat length (LB) is the wavelength-dependent optical parameter and is inversely related to the birefringence value. The high birefringence value of the sensor tends to the low beat length. The larger birefringence value, as well as lower beat length, is responsible for better polarization, preserving the performance of the sensor structure. There are many applications of the fiber structure that contain a beat length of less than 1 mm. Basically, the phase relationship between the orthogonally polarized modes can be represented through the beat length. The beat length for the proposed sensor was analyzed using Equation (Equation 5) and is plotted in Figure 5. It can be analyzed that the maximum beat lengths are 30.78 nm, 33.82 nm, and 37.04 nm at the operating wavelength 700 nm for the human teeth parts dentin, cementum, and enamel, respectively. The values are decreasing gradually with the increase of operating wavelength. At the wavelength 900 nm, the beat lengths reach the values of 15.12 nm, 16.06 nm, and 17.63 nm for the teeth parts, respectively.

### 5.4. Coupling Length (LC)

The LC is also defined as the distance for which the maximum guided power is transferred from one medium to the other medium. The values of LC can be determined using Equation (Equation 6) and are highlighted in Figure 6 for the different RI values of the human teeth parts. The coupling length also shows a downward curve for the increase of wavelength. The red, green, and blue lines specify the values of coupling length for the human teeth parts dentin, cementum, and enamel gradually.

### 5.5. Numerical Aperture (NA)

Basically, the values of NA depend on the RI values of the core and cladding regions. In this investigation, SF11 was used as the core material and the sensing analyte was kept as the cladding material. The RI values for the materials change with the variation of the operating wavelength. From the differences in these values, the numerical aperture can be attained. The curve of NA for the proposed sensor structure is shown in Figure 7. These values are gradually increasing within the wavelength range from 700 nm to 1000 nm. The values of NA 0.05, 0.04, and 0.02 RIU are noted for the human teeth parts at the wavelength 700 nm. In addition, the values of 0.09, 0.09, and 0.08 RIU are gained at the wavelength of 1000 nm. The relationship between the core and cladding materials is clearly visible through this optical parameter.

### 5.6. Power (P)

The output power spectrum mostly depends on the effective RI values of the sensor that change through the variation of the wavelength change and the fiber length. The power spectrum produces a sine curve and the values are saturated within 0 to 1 dB. The power spectrum for the proposed sensor structure is mentioned in Figure 8 for the sensing analytes. Maximum values of 0.93 dB, 0.95 dB, and 0.98 dB and minimum values of 0.09 dB, 0.009 dB, and 0.004 dB are achieved for the human teeth parts. From the wavelength 700 nm, the values are gradually decreasing, and after reaching the smallest value at the wavelengths 910 nm, 950 nm, and 990 nm, the curve starts increasing again. This sinusoidal curve shows better responses as the maximum value is so close to the value of 1 dB. The better response of the output power spectrum highly affects the values of the transmitted power spectrum or transmission spectrum. The power spectrum is the foremost optical parameter that has a great impact on the sensing performance of the simulated sensor structure. The relative sensitivity of the proposed sensor can be measured through the ratio of the core power and total power [34].

### 5.7. Transmission (T)

Another prominent optical parameter is the transmission spectrum. The values of the transmission characteristics are highly dependent on the output power spectrum. These values were gained from Equation (Equation 9) for the departure of the operating wavelength. The resulting transmission spectrum is shown in Figure 9. It shows that the transmission spectrum has a quasi-periodic spectrum that is measured by a resonance coupling between two polarized modes. The red, blue, and green lines indicate the transmission spectrum values for the analyte dentin, cementum, and enamel. The maximum peak values of −32 dB/m, −40 dB/m, and −48 dB/m were gained for the sensing analytes at the operating wavelengths 920 nm, 950 nm, and 990 nm. These values are needed for the calculation of sensing performance. As a result, the maximum values of the power spectrum tend to be the maximum values of transmission, and these values are responsible for obtaining the highest values of wavelength sensitivity (WS) responses.

### 5.8. Effective Mode Area (EMA)

The EMA is commonly used for calculating the effectiveness of fiber optic sensors. The value of EMA indicates the strength of the designed sensor. Normally, the fiber sensor modes have smooth transverse profiles [35]. The applied power has an inverse relationship with the EMA of the core region. This optical property is affected by the fundamental mode propagating the sensor structure. The EMA curve is shown in Figure 10 for the human teeth part of dentine within the working wavelength range 700 nm to 1000 nm. The curve is increasing linearly with the change in the operating wavelength. Moreover, the black line indicates the EMA for X polarization and the violet line specifies the EMA for Y polarization.

### 5.9. Wavelength Sensitivity (WS)

The wavelength sensitivity (WS) response can define the sensor efficiency and effectiveness for real-life applications. In this simulation, the values of WS were gained from the spectral peak values of the transmission spectrum. The values of WS vary with the variations of the sensing analytes. The WS responses are mentioned in Figure 11 for the two orthogonal polarized modes. It can be observed that the WS response is 28,482.43 nm/RIU for X polarization of the dentin, and this value is increasing and reaches the responses of 33,138.29 nm/RIU and 37,768.67 nm/RIU. Furthermore, the WS responses of 28,948.69 nm/RIU, 33,684.99 nm/RIU, and 38,396.56 nm/RIU are noted for the Y polarization of the sensing analytes.

### 5.10. Confinement Loss (CL)

The CL of the recommended sensor structure was measured using Equation (Equation 12) and is exhibited in Figure 12. The loss value specifies the scattering of the propagated light as well as the loss of sensor efficiency. The value of the confinement loss should be minimized so that the light propagates through the center and it should not be dispersed. These loss values can be determined on the basis of the diameter, pitch, and shape of holes. If the loss values become high, the sensor will be less effective. On the contrary, a low loss value indicates that the sensor will be more convenient. The provided sensor exhibits a very low loss value, as can be observed from the study of the simulated structure. The spectral loss peak values of 0.00015 dB/m, 0.00028 dB/m, and 0.00087 dB/m were obtained at the operating wavelengths 750 nm, 820 nm, and 955 nm for the sensing analytes dentine, cementum, and enamel, respectively.

Figure 13 mentions the phase-matching point between the true values of the core and spp modes and the corresponding confine loss values. It was shown that the real values of the core and spp mode intersect each other at the operating wavelength of 750 nm. Due to this, the confinement loss value increased to a maximum of 0.00014 dB/cm at a wavelength of 750 nm.

The overall optimal outcome analysis of the recommended sensor structure is listed in Table 2. In this work, a large number of fiber optic parameters were estimated and simulated. The values of the parameters that are responsible to obtain good responses are summarized in Table 2. The fiber optic parameters are highly related to each other. The high value of birefringence depends on the values of the effective RI of the sensor. In addition, the birefringence value has an inverse relationship with the beat and the coupling lengths. Moreover, the sensitivity response depends on the values of the transmission spectrum, and the transmission values are influenced by the output power values.

The best results from the suggested sensor were evaluated against prior outcomes. The comparison among different sensor structures and their responses is listed in Table 3.

## 6. Conclusions

Finally, a simulation was performed using Comsol Multiphysics software to design a sensor for the detection of human teeth disorders. The spectroscopy analysis for the human teeth parts of enamel, dentine, and cementum was performed to analyze the characteristics of the offered sensor structure. The RI values of the human teeth parts were used as the sensing analyte RI. The optical parameters value of power spectrum, transmission, wavelength sensitivity, and confinement losses are noted as 0.93 dB, −32 dB/m, 28,948.69 nm/RIU, and 0.00015 dB/m, respectively, for the dentine, 0.95 dB, −40 dB/m, 33,684.99 nm/RIU, and 0.00028 dB/m, respectively, for cementum, and 0.98 dB, −48 dB/m, 38,396.56 nm/RIU, and 0.00087 dB/m for enamel. In addition, through the spectroscopy-based analysis, the maximum relative sensitivity response of 99.99 was obtained. In essence, this is a recent approach to identifying problems with human teeth. Moreover, this technique is more efficient and less time-consuming compared to the existing methods. This study will be beneficial for future applications of the simulated sensor in the biosensing industry.

## Figures and Tables

**Figure 1 micromachines-14-01191-f001:**
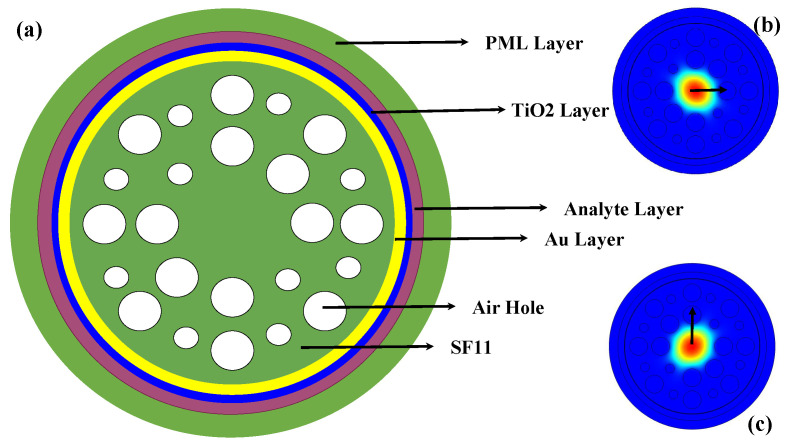
(**a**) Cross-sectional view of the simulated human teeth disease detection sensor structure, and the core mode for (**b**) X polarization and (**c**) Y polarization.

**Figure 2 micromachines-14-01191-f002:**
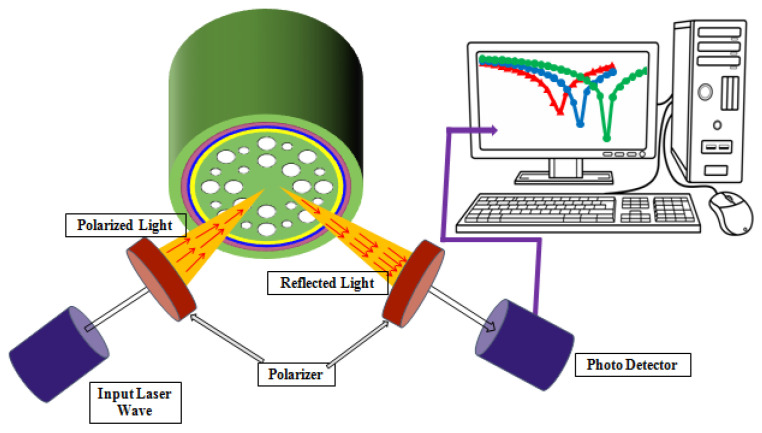
Recommended experimental setup diagram for the designed sensor structure.

**Figure 3 micromachines-14-01191-f003:**
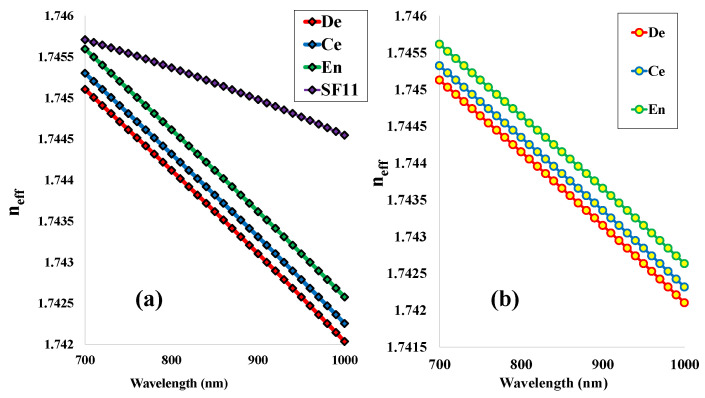
Refractive indices variation with corresponding wavelength variation for the tooth parts dentin, cementum, enamel, and the base material *SF11* for the orthogonal polarizations modes (**a**) for X polarization and (**b**) for Y polarization respectively.

**Figure 4 micromachines-14-01191-f004:**
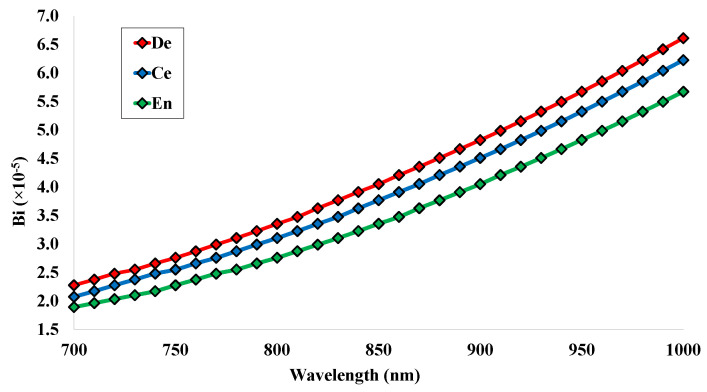
Birefringence values for the analyte of human teeth parts for the wavelength range 700 nm to 1000 nm.

**Figure 5 micromachines-14-01191-f005:**
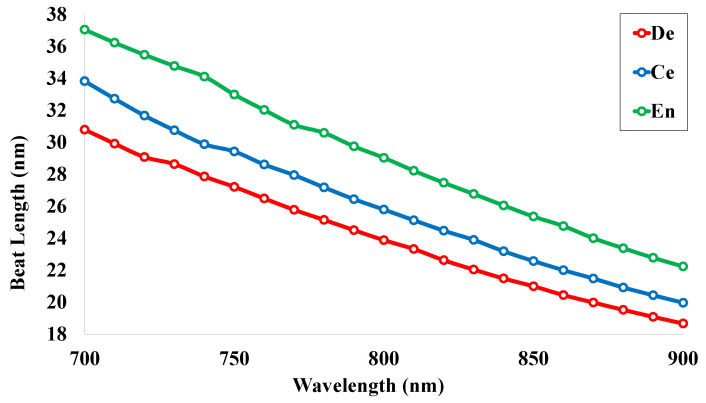
Beat length for the human teeth parts.

**Figure 6 micromachines-14-01191-f006:**
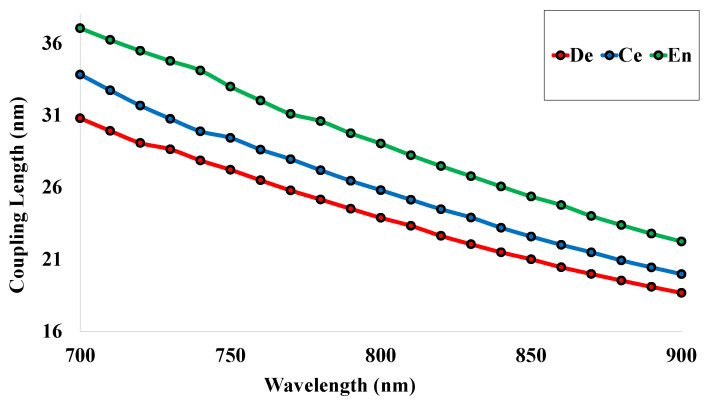
Coupling length for the human teeth affected parts.

**Figure 7 micromachines-14-01191-f007:**
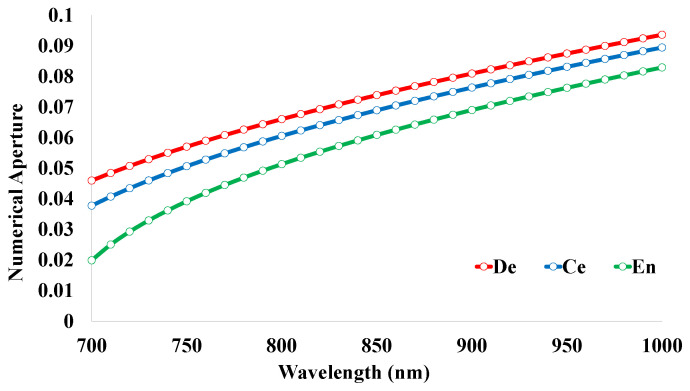
Numerical aperture for the proposed sensor structure.

**Figure 8 micromachines-14-01191-f008:**
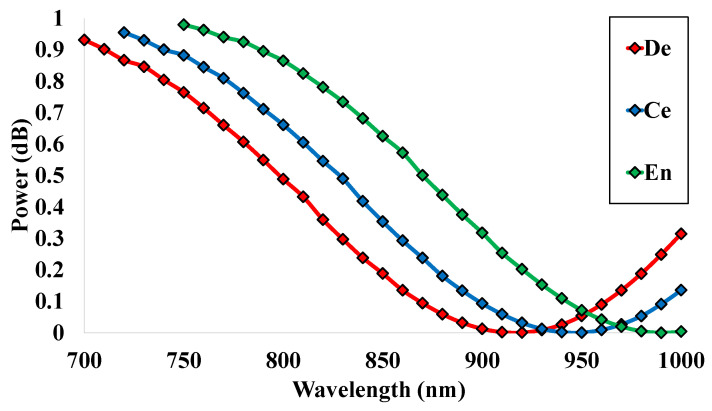
Output power spectrum for different analytes within the range 700 nm to 1000 nm.

**Figure 9 micromachines-14-01191-f009:**
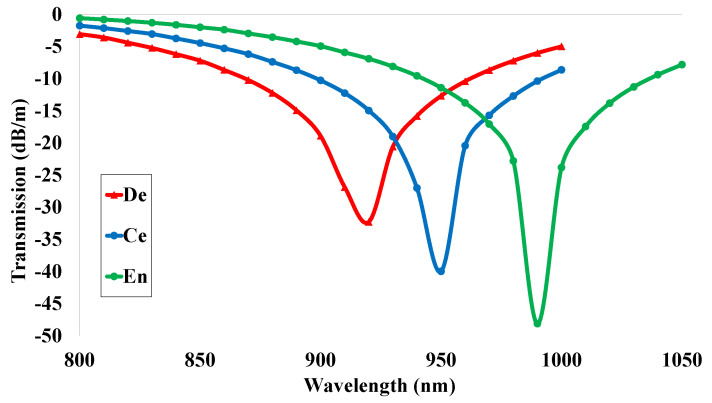
Transmission spectrum of the proposed sensor design within the operating wavelength 700 nm to 1000 nm.

**Figure 10 micromachines-14-01191-f010:**
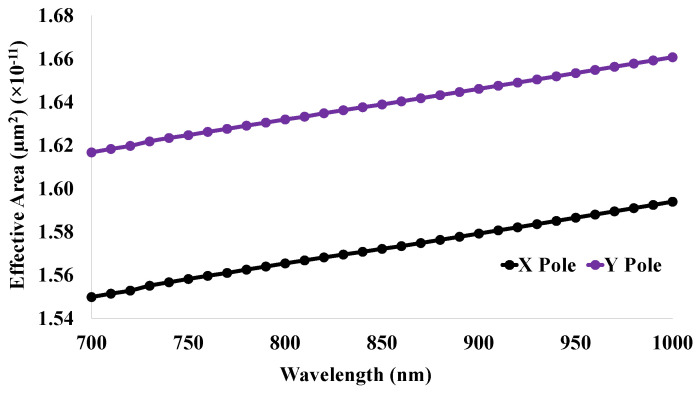
Effective mode area for the proposed sensor structure within the range 700 nm to 1000 nm both for X and Y polarizations for dentine.

**Figure 11 micromachines-14-01191-f011:**
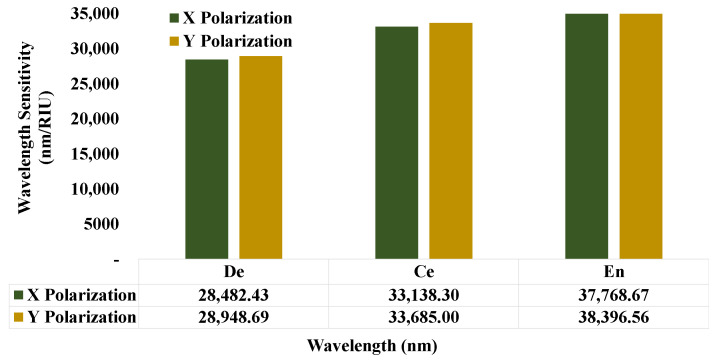
Wavelength sensitivity response for the analytes variation.

**Figure 12 micromachines-14-01191-f012:**
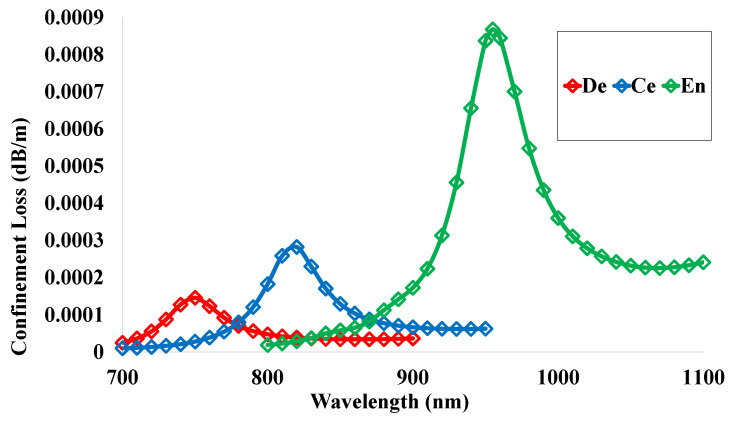
Confinement loss for the sensor structure range within 700 nm to 1000 nm.

**Figure 13 micromachines-14-01191-f013:**
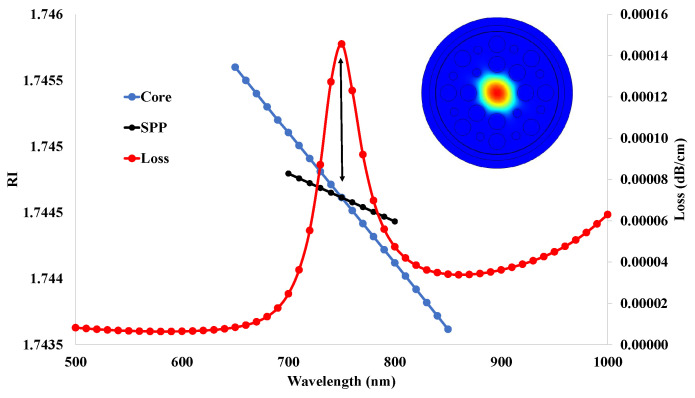
Phase-matching point of the core and spp mode.

**Table 1 micromachines-14-01191-t001:** Sellmeier constants for the material *SF11* [14].

Constants	Values	Constants	Values
$\alpha_{1}$	1.73759695	$\beta_{1}$	0.013188707
$\alpha_{2}$	0.313747346	$\beta_{2}$	0.0623068142
$\alpha_{3}$	1.89878101	$\beta_{3}$	155.23629

**Table 2 micromachines-14-01191-t002:** Optical performance analysis of the proposed sensor structure.

Sensing	Bi	LB	LC	NA	Pw	Tr	EMA	WS	CL
Analyte	(RIU)	(nm)	(nm)	(RIU)	(dB)	$\left(\frac{\mathrm{dB}}{m}\right)$	$\left(\mu m^{2}\right)$	$\left(\frac{\mathrm{nm}}{\mathrm{RIU}}\right)$	$\left(\frac{\mathrm{dB}}{m}\right)$
	$\times (10^{-5})$						$\left(\times 10^{-11}\right)$		
De	2.27	30.78	15.39	0.05	0.93	−32	1.55	28,948.69	0.00015
Ce	2.07	33.82	16.91	0.04	0.95	−40	1.55	33,684.99	0.00028
En	1.89	37.04	18.52	0.02	0.98	−48	1.55	38,396.56	0.00087

**Table 3 micromachines-14-01191-t003:** Comparison of different optical responses with previous sensor analysis.

Structure	Bi (RIU)	WS (nm/RIU)	RS (%)	CL (dB/m)	Ref.
Triple-layer SPR-based sensor	3.59 × $10^{-4}$	28,046.205	——	3.5 × $10^{4}$	[36]
High birefringence gas sensor	8.76 × $10^{-3}$	——	96.7	0.11	[37]
Sulfur dioxide gas sensor	1.0 × $10^{-3}$	——	70	0.000136525	[13]
PCF-based liquid sensor	8 × $10^{-3}$	——	49.13	5.583 × $10^{-5}$	[38]
D-shape SPR-based RI sensor	——	31,000	——	474.32	[39]
SPR-based biosensor	——	34,000	——	0.79 × $10^{-2}$	[40]
D-Shaped germanium-doped PCF sensor	1.06 × $10^{-2}$	5600	——	0.00680	[18]
Human teeth disease detection sensor	1.89 × $10^{-5}$	38,396.56	99.99	0.00087	This work

## Data Availability

The data that support the finding of this study are available from the corresponding author upon reasonable request.

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
