# Peer review of "Au-TiO2-Coated Spectroscopy-Based Human Teeth Disorder Detection Sensor: Design and Quantitative Analysis"

_micromachines, 2023, doi:10.3390/mi14061191_

Round 1

Reviewer 1 Report

The authors proposed a SPR-PCF sensor for human tooth disease detection using FEM simulation. This work has significant potential for sensor design and real-time temperature remote sensing applications. Some revisions are needed to publish this manuscript.

1.      Please elucidate the novelty of their work in the introduction section.

2.      At what wavelength is the E-field distribution in Figures (b) and (c) plotted? It should be mentioned.

3.      With respect to Figs. (b) and (c), if possible, I suggest providing the real part of effective RI versus wavelength of the core-guided mode and the SPP mode (authors can refer to the related articles).

4.      For readers to follow, the values of angular frequency ω, damping frequency γD, permittivity ϵα, plasmonic frequency ωD, and the weighting factor Δϵ that used in Equation 2 should be mentioned in the text.

5.      Typos, e.g., line 176. In the same manner, please check them throughout this manuscript.

6.      It is helpful to include a reference (doi: 10.2528/PIERB10042405) on PCF birefringence.

7.      If possible, please elaborate in more detail the mechanism presented in Figs. 9 and12.

8.      For readers to know the other approaches for the design of the SPR-PCF sensor, a reference ( https://doi.org/10.3390/photonics9120916) is suggested in the text.

Reviewer 2 Report

This work describes using Comsol Multiphysics software to design a sensor for the detection of human teeth disease. The spectroscopy analysis for the human teeth parts of enamel, dentine, and cementum has been performed to analyze the characteristics of the offered sensor structure. Below are some questions and the detail revision points for the authors' consideration.

1. In Introduction, the progress of a human teeth disease detection and spectroscopy-based analysis isnt introduced. Need to improve.

2. In Introduction, page 32-47, please check the logical relation, this part needs to show some progress, but there are problems with the logic and content. Please revise.

3. Where is the description of the experimental data measured by the experimental device in Figure 2, and how does this relate to the theoretical calculation?

  • There is a problem with the logical description. I hope I can sort it out to make the paper clearer

Reviewer 3 Report

This work proposes an optical sensor for human teeth disease detection. The manuscript is well organized and presented. However, all the data is obtained from simulation. What's the accuracy compared with experiment?

The introduction part should also be improved to emphasize on the importance and novelty of this work. 

NA

Round 2

Reviewer 2 Report

  • This edition is fit for publication

Reviewer 3 Report

Thanks the authors to clearly state that this work is for simulation only. The introduction part is also well improved. 

All my questions have been well addressed. 

NA